# Application of Social Big Data to Identify Trends of School Bullying Forms in South Korea

**DOI:** 10.3390/ijerph16142596

**Published:** 2019-07-21

**Authors:** Hayoung Kim, Yoonsun Han, Juyoung Song, Tae Min Song

**Affiliations:** 1Department of Child Psychology and Education, Sungkyunkwan University, Seoul 03063, Korea; 2Department of Social Welfare, Seoul National University, Seoul 08826, Korea; 3Department of Administration of Justice, Pennsylvania State University, Schuylkill Haven, PA 17972, USA; 4Department of Health Management, Sahmyook University, Seoul 01795, Korea

**Keywords:** school bullying forms, social big data, TF-IDF, Future Signals, South Korea

## Abstract

As the contemporary phenomenon of school bullying has become more widespread, diverse, and frequent among adolescents in Korea, social big data may offer a new methodological paradigm for understanding the trends of school bullying in the digital era. This study identified Term Frequency-Inverse Document Frequency (TF-IDF) and Future Signals of 177 school bullying forms to understand the current and future bullying experiences of adolescents from 436,508 web documents collected between 1 January 2013, and 31 December 2017. In social big data, sexual bullying rapidly increased, and physical and cyber bullying had high frequency with a high rate of growth. School bullying forms, such as “group assault” and “sexual harassment”, appeared as Weak Signals, and “cyber bullying” was a Strong Signal. Findings considering five school bullying forms (verbal, physical, relational, sexual, and cyber bullying) are valuable for developing insights into the burgeoning phenomenon of school bullying.

## 1. Introduction

The changing pace of school bullying is as rapid as the pace of modern society, making it difficult to predict the evolving forms used in school bullying. School bullying is a public health concern in South Korea (hereafter Korea), in particular, which has experienced remarkable changes in school bullying forms. As stated in a recent national report, the most common forms of bullying victimization in Korea among elementary school to high school students (primarily between the ages of 7 and 18) were verbal aggression (34.7%) and group ostracism (17.2%), followed by stalking (11.8%), cyber (10.8%), and physical (10.0%) bullying [1]. From 2014 to 2017, there has been a decreasing trend in verbal and relational bullying but a relative increase in sexual bullying, cyber bullying, and other unclassified forms of bullying [2]. More diverse and complex forms of school bullying have been observed, as digital native adolescents spend a typical day with peers in both offline and online environments and peer interactions are no longer limited by time and space [3]. When digital native bullies assault victims physically or sexually offline, for example, they also broadcast the live bullying scene or post pictures of the victim’s appalling injuries on social network services (SNS) to mock the victim as a form of secondary attack.

As school bullying is vast, diverse, and frequent, it is not easy to trace all scenes of school bullying through traditional social survey methods that are limited in sample size, number of questionnaire items, and data collection time points. From this perspective, many social scientists emphasize that big data can no longer be overlooked, as it offers a new paradigm for social science methodology [4]. Social big data may be an alternative tool for research on the school bullying experience of digital native adolescents by having the advantage of detecting changes in patterns of school bullying more broadly, diversely, and quickly than traditional survey-based data. Therefore, using social big data, the current study examined keywords of school bullying forms and identified relevant Future Signals.

### 1.1. School Bullying and Big Data

Big data is defined as “extremely large data sets that may be analyzed computationally to reveal patterns, trends, and associations, especially relating to human behavior and interactions” [5]. Big data is often compared to traditional data in terms of its distinctive characteristics, namely the “3Vs,” which are volume (the vast scale of the data), variety (the breadth of types of data), and velocity (the rapid generation of data) [6]. A high volume of data provides abundant information on the research subject; variety allows for the consideration of new information beyond the researcher’s initial study design or anticipated response, and high velocity has the advantage of allowing researchers to trace individual behaviors by the second. The 3Vs of big data correspond to characteristics of the contemporary phenomenon of school bullying among adolescents, which has become more widespread (volume), diverse (variety), and frequent (velocity).

#### 1.1.1. Volume of School Bullying

School bullying incidents referred to the Autonomous Committees for Countermeasures against School Violence increased from 17,749 to 23,673 between 2013 and 2016 [7]. The fact that Korean teenagers spend much of their time online perhaps accounts for the increase of school bullying. Of every 10 youths in Korea, 9 own smartphones, 8 use their smartphones daily, and 7 think that smartphones are indispensable to everyday life [8]. The traditional victim–bully dynamic involved individuals in the school class, whereas nowadays the availability of online social networking has allowed the scope of bullying behavior to expand to the whole school, the entire community, and even peer groups separated by thousands of miles.

#### 1.1.2. Variety of School Bullying

School bullying has evolved into diverse and new forms that span online and offline domains. Adolescents constantly force victims to enter a group chatroom despite knowing that the victims do not want to be invited (called *ka-tok jail*, or “prison chatroom”), or send attacking or abusive messages on group-based SNS (*tte-ka*, or “message sent by a mob”) [9]. Furthermore, traditional bullying among digital natives often continues online after school and on the weekends, causing secondary harm to the victim [10]; adolescents can harass victims using online devices even when they are offline, and conflicts online can motivate traditional bullying in school.

#### 1.1.3. Velocity of School Bullying

Present-day adolescents are more frequently exposed to school bullying than in the past. According to the school violence survey of the Korean Ministry of Education from 2012 to 2014 [11], the percentage of victims who experienced school bullying “once or twice in six months” decreased from 62.5% to 47%, while the percentage of victims who experienced school bullying “almost every day” increased from 9.1% to 17.16%. As digital native adolescents have 1.5 times more friends on SNS than adults [12], they can easily send hostile messages to victims and spread victims’ personal information or secrets to other peers during class or break time, and even at midnight.

### 1.2. Forms of School Bullying

Traditionally, school bullying has been categorized as verbal bullying, physical bullying, relational bullying, or sexual bullying [13]. With greater youth engagement in the online domain, cyber bullying may be considered as another type of school bullying. All five types of school bullying have unique characteristics and specific forms used to harass the victim.

#### 1.2.1. Verbal Bullying

Verbal bullying refers to the act of expressing certain words or verbal utterances to offend others, and includes specific forms such as swearing, laughing, teasing, threatening, and cursing. Verbal bullying is the most common type of bullying in Korea, as well as in many other countries [14]. A unique feature of verbal bullying is that it is less restrictive in terms of the physical strength of the actors and place of occurrence, such as online and offline domains. Sometimes verbal aggression may not be perceived by the perpetrator or even the victim as bullying behavior, as the harm it induces is not as visible as that of other types of bullying [15].

#### 1.2.2. Physical Bullying

Physical bullying refers to the act of physically distressing another person. Common forms of physical bullying include not only injuring the victim by hitting, pushing, grabbing, kicking, throwing objects, tripping, or scratching, but also placing the victim in captivity or taking and destroying their property [16]. Physical bullying is commonly perpetrated by males, but the number of female group assaults has been increasing recently [17]. Physical violence may lead to the most severe injuries among both victims and bullies, and adolescents who engage in physical harm may escalate to committing juvenile crimes [18].

#### 1.2.3. Relational Bullying

Forms of relational bullying include ignoring or excluding someone and spreading rumors or sharing personal secrets that can harm a person’s reputation or relationship with others. Relational bullying is often done in combination with other types, such as verbal or physical bullying, and it may not be visible to external observers, especially when executed indirectly. Furthermore, some perpetrators may not recognize that they are actual perpetrators of bullying, which makes it difficult to quantify relational bullying on a larger scale. Relational bullying is often done in groups rather than by individuals, and can have a continuous negative effect on social adjustment and interactions with others even after adolescence [19].

#### 1.2.4. Sexual Bullying

Sexual bullying refers to violent behavior intended to harm or humiliate a person sexually, whether physically or not. It encompasses bullying forms such as sexual assault or harassment behavior by individuals or groups, as well as forcing conditional sex or prostitution for monetary extortion. Sexual bullying can have severe psychological impact to victims, such as somatization symptoms, anxiety, and depression [20]. Victims of sexual harassment have a stronger tendency to be hesitant about reporting their experience or not to report it at all than do victims of other types of bullying [21]. They are also more likely to be exposed to repeated sexual violence or to develop serious psychological trauma [22].

#### 1.2.5. Cyber Bullying

All bullying behavior in the online domain using an electronic medium, such as a mobile phone, computer, or other private devices, is regarded as cyber bullying [23]. There are diverse cyber bullying types, such as flaming, harassment, denigration, impersonation, outing and trickery, exclusion, and cyberstalking [24]. Cyber bullying is a prevalent and ever-growing problem among digital native adolescents, whereas traditional bullying seems to be declining [25]. Cyber bullying has the characteristics of intention, repetition, and power imbalance (as in traditional bullying), along with two additional characteristics: publicity and anonymity. In online space, adolescents can insult a victim publicly and anonymously [26].

### 1.3. The Current Study

Although big data may present promising opportunities to understand the rapid change in the forms of school bullying, big data research on school bullying has rarely been conducted in Korea compared to other research areas related to psychological adjustment and risk behaviors, such as happiness, depression, and suicide [27,28,29]. Even the few big data studies on school bullying that do exist are limited in dealing only with cyber bullying and not school bullying in general [30].

To address the limitations in the previous research, this study attempted to suggest trends of school bullying forms in Korea using social big data. Answering the following research questions, this study aimed to describe the present state of school bullying forms and address the further insights for forthcoming school bullying issues: (1) What are the important keywords of school bullying forms in social big data between 2013 and 2017? (2) What are the Future Signals of school bullying forms as detected by social big data?

## 2. Materials and Methods

### 2.1. Materials

This study analyzed 436,508 web documents mentioning terms related to “school bullying” between 1 January 2013, and 31 December 2017, on 279 online channels. The data were collected from blogs (e.g., Naver, Daum, Tistory, Egloos), SNS (e.g., Twitter), social networking websites (e.g., YouTube, Naver KnowlegeiN, Nate talk), and 257 news sites—all of which are widely used in Korea. The data were collected and organized by SKT Smart Insight Cooperation, a leading Korean telecommunications company. This company used a web crawler, which is a set of programs used to search and gather data from web-based content. Articles, comments, tweets, and news were collected, and these documents contained a total of 177 terms regarding specific forms of school bullying (refer to Table A1).

In this paper, words collected from social big data are cited with double quotes (“”). The pronunciations of some words that can only be expressed in Korean are shown in italics without translation. As “school bullying” is a comprehensive concept that covers all terms and all types, this term was excluded from interpretation of results. In the phase of Term Frequency-Inverse Document Frequency (TF-IDF) and Future Signals analysis, rarely used words, which illustrate a few particular cases, were excluded from the analyses because they are likely to affect the average frequency and change rate [31]. Only the top 30 words in the TF-IDF ranking of the 5-year period were applied among all 177 words related to school bullying forms in this study.

### 2.2. Analytic Methods

To identify longitudinal changes and important signals of school bullying forms, the following methods were used: Term Frequency-Inverse Document Frequency (TF-IDF) and Future Signals analysis. All analyses were conducted using R 3.4.3 (R Foundation for Statistical Computing, Vienna, Austria).

#### 2.2.1. Term Frequency-Inverse Document Frequency (TF-IDF)

The TF-IDF is the most common method in big data research for text mining and an indicator of the importance of words by investigating both the frequency of words in the entire document (Term Frequency (*TF*)) and the frequency of the documents in which the words appeared (Document Frequency (*DF*)). The Inverse Document Frequency (*IDF*) is obtained by inverting the *DF* and then taking the log in order to express it as a value between 0 and 1. The *TF* represents the visibility of the word, the DF points to the diffusion degree of the word, and TF-IDF is an index representing the importance of the word.
(1)$$TF_{x,y}=\frac{\mathrm{The}\;\mathrm{number}\;\mathrm{of}\;\mathrm{term}\;\left(x\right)\;\mathrm{in}\;a\;\mathrm{document}\;\left(y\right)}{\mathrm{The}\;\mathrm{number}\;\mathrm{of}\;\mathrm{total}\;\mathrm{terms}\;\mathrm{in}\;a\;\mathrm{document}\;\left(y\right)}$$
(2)$$DF_{x}=\frac{\mathrm{The}\;\mathrm{number}\;\mathrm{of}\;\mathrm{documents}\;\mathrm{containing}\;\mathrm{term}\;\left(x\right)}{\mathrm{Total}\;\mathrm{number}\;\mathrm{of}\;\mathrm{documents}}$$
(3)$$IDF_{x}=\log\left(\frac{1}{DF_{x}}\right)$$
(4)TF−IDF=TF×IDF

#### 2.2.2. Future Signals

Future Signals refer to currently imprecise and non-mainstream signals with a potential to change in the future. This term was originally referred to as a “weak signal” by Ansoff [32], and referred to a small noise or symptom with a pattern of future change. Hiltunen stated that the “future sign clarifies the difference between what is really happening (issue) and what its information value (signal) is”, and highlighted its worth in predicting the future [33].

To find Future Signals, two change rates—Degree of Visibility and Degree of Diffusion—are calculated, based on the results of TF-IDF analysis [34]. Degree of Visibility (*DoV*) indicates how much a word is used over time, and Degree of Diffusion (*DoD*) shows how the spread of the word across different documents varies over time. The following equations show how to evaluate the *DoV* or *DoD* of word *i* in period *j*. Here, *NN* is the total number of documents, *n* is the whole period of time, and *tw* is a time weight; previous researchers have usually used 0.5 for *tw* [31].
(5)$$DoV_{ij}=\left(\frac{TF_{ij}}{NN_{j}}\right)\times \left\{1-tw\times \left(n-j\right)\right\}$$
(6)$$DoD_{ij}=\left(\frac{DF_{ij}}{NN_{j}}\right)\times \left\{1-tw\times \left(n-j\right)\right\}$$

Based on the previously obtained *TF*, *DF*, *DoV*, and *DoD*, two maps—the Keyword Emergence Map (KEM) and Keyword Issue Map (KIM)—are generated, as depicted in Figure 1. The vertical line refers to the average frequency (*TF* or *DF*), and the horizontal line the average change rate (*DoD* or *DoV*). Each quadrant contains different information about words in terms of the present and future. Terms that were detected as a common signal in the two maps of KEM and KIM were defined by their signal type. In the Weak Signal quadrant, words have a lower current frequency but higher rate of growth, which suggests that they may increase quickly in the future. In the Strong Signal (also known as a trend) quadrant, words have high frequency and a high increasing rate. The third quadrant, called Latent Signal, is characterized by words with low frequency and change rates. The fourth quadrant, Strong but Low Increasing Signal, comprises words with a high frequency but low change rates.

## 3. Results

### 3.1. Descriptions of Data

During the five-year period, documents describing school bullying were relatively stable between the beginning of 2013 and middle of 2016, but surged from the middle of 2016 to the end of 2017, with peaks observed in December 2016, June and July 2017, September 2017, and December 2017 (Figure 2). The peak in December 2016 represents the online movements to eradicate cyber bullying and sexual harassment. These two forms of school bullying were consistently mentioned as part of a fandom activity centered on the hash tags “cyber bullying”, “sexual assault”, and “vicious sexual harassment”. The second case is related to school bullying at an elementary school in Seoul. In this case, several elementary school students brutally assaulted another student using “body wash” and a “blanket”, which became a major social controversy in June and July, 2017. In September 2017, there was group-based violence among teenagers in the cities of Busan, Gangneung, and Cheonan that received considerable attention among social network sites and news channels. Lastly, in December 2017, there was another severe group violence that occurred in Gwangmyeong City, as well as renewed increases in fandom activity on Twitter related to the issue of cyber bullying.

### 3.2. Term Frequency-Inverse Document Frequency (TF-IDF)

As a result of TF-IDF, different trends emerged between different forms of bullying, namely, physical, relational, sexual, and cyber bullying (Table 1). The exception, however, was verbal bullying: “verbal violence” was the only word in the top 30, and there have been few noticeable keywords or changes in verbal bullying.

Words for physical bullying such as “assault” and “violence” regularly ranked at the top. There were words that describe unique forms of physical bullying, such as “*bang-shuttle*” and “blanket”, as well as words that are used to express the body in a neutral manner, not words limited to school bullying behavior, such as “head”, “fist”, and “mark”.

The word “*wang-dda*”, which is a representative relational bullying word in Korean, was repeatedly located at the top of TF-IDF, and “bullying” seemed to emerge gradually as the main keyword. In contrast, the importance of “ostracize”, “group ostracize”, “rumor”, and “alienation” were decreased with time.

The most noticeable increase of importance over time was in sexual bullying words. Words related to sexual bullying appeared throughout the entire period, but particularly in the most recent 2 years “sexual harassment” and “vicious sexual harassment” had high rankings in TF-IDF.

Diverse cyber bullying-related words emerged steadily as key terms of school bullying behavior over the 5 years. There are terms associated with attacking others directly and publicly, such as “malicious comments”, to blocking access by others in social networking channels, such as “block” and “block list”, and to using digital devices, such as “video-clip”, “image”, “text message”, “mobile phone”, “filming”, and “capture”.

### 3.3. Future Signals

Every form of school bullying was verified as a different signal (Table 2): verbal bullying as Latent Signals (e.g., “verbal violence”), physical bullying as Weak (e.g., “group assault”) and Strong but Low Increasing Signals (e.g., “violence”), relational bullying as Strong but Low Increasing Signals (e.g., “*wang-dda*”), sexual bullying as Weak Signals (e.g., “sexual harassment”), and cyber bullying as Strong Signals (e.g., “cyber bullying”).

With the words “group assault” and “sexual harassment”, the major recent issues of school bullying were also captured as Weak Signals. The word “group assault” demonstrates the well-known group bullying cases of Busan, Gangneung, Cheonan, and Gwangmyeong in the latter half of 2017. Reflecting the problem of sexual harassment in fandom, “sexual harassment” was also detected as a Weak Signal. With a high rate of increase compared to other words, they have a high likelihood to become more socially impactful words with respect to school bullying behavior in the future.

“Cyber bullying” emerged as a Strong Signal of school bullying. In this result, school bullying issues that were mentioned previously are reflected. There was a movement to eliminate cyber bullying within fandom in December 2016 and December 2017, which spread widely through Twitter.

Most words were defined as Latent Signals (e.g., “perpetration”, “harassment”, “ostracize”), among the top 30 words of the TF-IDF over the 5-year period. As keywords of school bullying in social big data, they are less important than other signal words, because they had relatively low frequency and change rate across the documents.

Lastly, three words—“violence”, “*wang-dda*”, and “school bullying”—were identified as Strong but Low Increasing Signals. Each word refers to a different form of school bullying, except for “school bullying”, with “violence” for physical bullying and “wang-*dda*” for relational bullying. These words were high in frequency but did not show a high rate of change. It implies that they are words generally used to describe school bullying behavior throughout most of the period.

## 4. Discussion

School bullying has long been a persistent problem, in spite of growing research and social attention. This study sought to identify the trends of school bullying forms among digital native adolescents in Korea over the span of five years (2013–2017) using social big data. The results of TF-IDF and Future Signals analysis were derived from the frequency and diffusion of words, and provided noticeable keywords and signals of school bullying forms in Korea that were mentioned in over 400,000 online documents. The key findings of the current study are discussed in the following sections.

### 4.1. Trends of School Bullying Forms in Korea

First, sexual bullying showed a surge in word usage from 2013 to 2017. According to TF-IDF results, occurrence of sexual bullying-related words increased rapidly in the last two years, and the word “sexual harassment” was detected as a Weak Signal. These results suggest that adolescents’ sexual bullying issues may increasingly become the center of social interest. In fact, it is easier to create or share sexual content using private digital device such as smartphones than in the past. For these reasons, previous studies emphasized that teenagers’ sexual experiences, including sexting, have been increasing recently, and these experiences in turn have resulted in a surge in sexual violence behavior [35].

Second, words concerning physical and cyber bullying forms in social big data have been high in frequency and growth rate over the past five years. For example, “group assault” was detected as a Weak Signal, and “cyber bullying” was identified as a Strong Signal. These results depict the bullying situation in Korea, where physical bullying forms are perceived as extremely serious among youth [36]. The recent surge in group assault incidences by adolescents and physical violence cases, particularly among female students in Korea, have also been indicated using traditional social science research methods [17,37]. Equally importantly, cyber bullying has attracted much attention as a major social problem, especially when considering digital native adolescents whose activity domains include both online and offline spaces [38]. The results suggest that the problem of cyber bullying is not limited to the school grounds, but has spread to other environments, such as the fandom communities of teenagers.

Third, relational bullying-related words have consistently appeared in top keywords in 2013–2017. The importance and spread rate of words describing relational bullying are mostly stable or slightly decreasing, as indicated by the TF-IDF and Future Signals results. This trend of relational bullying in social big data confirms that relational bullying has remained stable in Korea, in accordance with the results of national surveys [2]. However, independent relational bullying behaviors may have been observed less frequently in social big data, as relational bullying has the tendency to emerge in subtle and complex ways by combining physical and cyber bullying in recent years [39,40].

Fourth, forms of verbal bullying were not explicitly represented in social big data, because generally verbal bullying is more likely to be noticed in the overall context of conversations rather than in terms of direct forms of verbal expressions. Moreover, when verbal bullying is conducted together with other forms (e.g., verbal violence in a scene of physical violence, sexual harassment through spoken words), verbal bullying forms are less likely to be specifically mentioned in the document when describing the bullying scene, and thus such terms are more likely to be submerged than other bullying forms. For these reasons, it may be difficult to capture texts referring to written–verbal bullying behaviors [41] in social big data; thus, identifying alternative methods to grasp the situational context of verbal violence is warranted in future research.

### 4.2. Implications in School Bullying Research

The strength of the current study is that it covered three distinctive characteristics—volume, velocity, and variety—of the contemporary school bullying problem using social big data analysis. First, comprehensive insight into school bullying forms across all five school bullying forms was possible based on the abundant information in the data. In this study, about 88,000 documents per year and 177 terms related to school bullying were collected. The number of study respondents, however, have been limited in research conducted by social survey data; 63% of school bullying articles were analyzed with under 1000 respondents [42], and 81% of cyber bullying articles were examined with under 1000 respondents, with the maximum number of respondents being 11,964 [43]. Furthermore, in analyzing big data, the current study was able to cover a number of bullying forms used by digital natives, and numerous school bullying situations represented in Korean society.

Second, the results of this study may encompass diverse and unpredictable school bullying circumstances among digital natives through exploratory data collection and analysis. Previous research studies focused on only a few existing forms of school bullying or the experience of school bullying, and a handful of school bullying experiences were collected by previous researchers with a few questions [44,45]. In this study, nationally recognized incidents of severe group assaults, sudden movements on cyber bullying in fandom, and new bullying forms, such as using body wash, were additionally included, all of which may not have been anticipated in the initial construction stage of research. The results of this study encompass evolving forms of school bullying, and additionally have an advantage of tracking the experience of school bullying among digital native adolescents.

Third, real-time data of this study compensated for the limitation of discontinuity in traditional data, and the results, therefore, may provide elaborative information on the school bullying problem. Previous research studies have mostly explained a single momentary experience and have not been able to explain continuous occurrence of school bullying behaviors [37], and hence were limited in suggesting how digital native adolescents’ violence forms are rapidly changing or what types of bullying are noticeably increasing. However, in this study, continuous scenes of school bullying were accumulated every hour and every minute over five years. It is critical that the analysis, which was based on precise temporal information, presented Future Signals that should be considered with regard to the school bullying phenomenon.

### 4.3. Limitations

The results of this study, however, need to be considered in the context of the following limitations. First, documents describing the same social issue may be generated by many people and identical documents may be replicated through re-postings of the original documents. Second, careful interpretation and application is recommended when suggesting implications for intervention using these results at the individual level. Since social big data, such as web-based text data, contains reactions or opinions about school bullying as well as experiences of it, the results of this paper are more suitable for offering macro perspectives on such issues, such as the establishment of public policies and preventive strategies at the social group level. Third, because of the privacy protection regulation, it was not possible to identify the characteristics of unique individuals in this study. Additional research is warranted to determine the causes of diverse school bullying forms and how these forms effect adolescents’ everyday life.

## 5. Conclusions

This study described the present state of school bullying forms in South Korea by examining the frequency and rate of increase of words representing bullying forms in social big data collected over five years. Findings recommend developing a monitoring system and countermeasures for emerging school bullying forms, especially related to sexual, physical, and cyber bullying. The current trends of school bullying forms depicted in this study are valuable to understand digital natives’ school bullying behavior and to establish advanced prevention strategies for school bullying issues.

## Figures and Tables

**Figure 1 ijerph-16-02596-f001:**
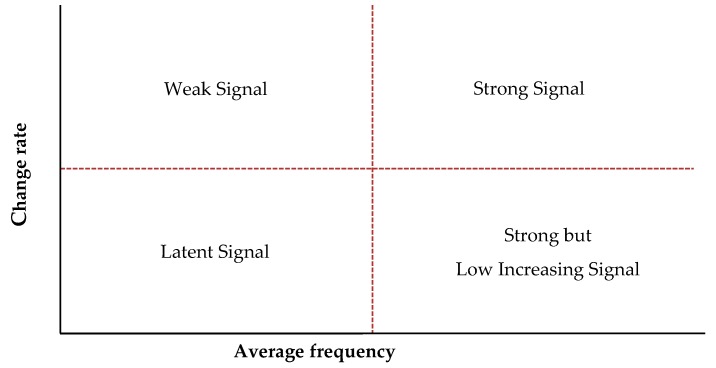
Map of Future Signals.

**Figure 2 ijerph-16-02596-f002:**
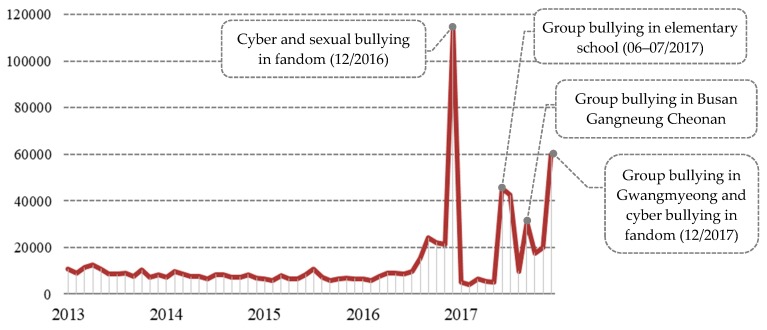
Web-document occurrence from 2013 to 2017.

**Table 1 ijerph-16-02596-t001:** Top 30 words as determined by Term Frequency-Inverse Document Frequency (TF-IDF).

Rank	2013	2014	2015	2016	2017
1	school bullying	school bullying	*wang-dda*	cyber bullying	school bullying
2	*wang-dda*	*wang-dda*	school bullying	*wang-dda*	assault
3	violence	violence	assault	school bullying	bullying
4	assault	ostracize	violence	sexual harassment	violence
5	group ostracize	head	ostracize	violence	mark
6	cyber violence	cyber violence	cyber violence	capture	cyber bullying
7	ostracize	group ostracize	head	mobile phone	sexual harassment
8	head	assault	harassment	cyber violence	*wang-dda*
9	harassment	mobile phone	group ostracize	vicious sexual harassment	blanket
10	mobile phone	harassment	mobile phone	assault	mobile phone
11	crime	*bang-shuttle*	group assault	bullying	crime
12	*bang-shuttle*	crime	crime	ostracize	group violence
13	cyber	cyber	image	head	group assault
14	image	fist	video-clip	group ostracize	harassment
15	video-clip	verbal violence	cyber	crime	head
16	group harassment	video-clip	capture	harassment	ostracize
17	rumor	image	*bang-shuttle*	stalking	cyber violence
18	filming	rumor	rumor	verbal violence	image
19	sexual violence	text message	filming	targeted	malicious comments
20	group assault	*eun-dda*	perpetration	perpetration	perpetration
21	perpetration	perpetration	text message	block	group ostracize
22	verbal violence	group harassment	verbal violence	image	*eun-dda*
23	text message	murder	sexual violence	cyber	student assault
24	sexual assault	group assault	assaulting	killing	video-clip
25	*eun-dda*	malicious comments	group harassment	sexual violence	sexual violence
26	malicious comments	sexual violence	*eun-dda*	abuse	cyber
27	capture	cyber bullying	abuse	rumor	stalking
28	murder	abuse	fist	video-clip	student violence
29	alienation	filming	alienation	*bang-shuttle*	text message
30	fist	alienation	teasing	block list	rumor

Note: *bang-shuttle* = forcing to deliver bread; *eun-dda* = ambiguous bullying in Korean; *wang-dda* = bullying in Korean. Most words directly describe forms of school bullying, however, some neutral words indicate the tools used in bullying events (e.g., blanket, image, mobile phone, text message, and video-clip); some indicate exclusion behavior in social network services (SNS) (e.g., block, block list); some indicate teasing behavior using digital devices (e.g., capture, filming); some indicate the result of bullying behavior (e.g., mark); others are detailed description of bullying behavior (e.g., cyber, targeted). The pronunciations of some words that can only be expressed in Korean are shown in italics without translation.

**Table 2 ijerph-16-02596-t002:** Future Signals of school bullying forms.

Weak Signals	Strong Signals
group assault, sexual harassment	cyber bullying
**Latent Signals**	**Strong but** **Low Increasing Signals**
perpetration, harassment, video-clip, ostracize, head, text message, crime, *bang-shuttle*, cyber, cyber violence, sexual violence, rumor, stalking, malicious comments, verbal violence, image, *eun-dda*, mobile phone, group harassment, group ostracize, filming	*wang-dda*, violence, school bullying

The pronunciations of some words that can only be expressed in Korean are shown in italics without translation.

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
