# Peer review of "Application of Social Big Data to Identify Trends of School Bullying Forms in South Korea"

_ijerph, 2019, doi:10.3390/ijerph16142596_

Round 1

Reviewer 1 Report

I recommended moving forward with the article.

I particularly liked their inclusion of the 3V's and creative use of data.

I recommended removing figures 3 and 4

Author Response

Response to Reviewer #1

1.     I recommended removing figures 3 and 4.

è The authors thank the Reviewer for this suggestion. We have removed Figures 3 and 4.

Reviewer 2 Report

The manuscript shows social big data methods application as alternative tools for research on the school bullying phenomenon. The study describes the state of school bullying in the South Korea by examining the frequency and rate of increase of words representing school bullying in social big data collected from over 400,000 web documents over five years (2013-17). The results of this social big data analysis gives comprehensive insight into the school bullying forms based on abundant information represented in Korean schools.

But the description of the selected forms of school bullying forms is not sufficient. It would be helpful to explain more the forms (methods)  not related by meaning to the school bullying.

In Table 2 the forms (methods) of school bullying should be better described as related to school bullying, such as:  video-clip, head, text message, cyber, image, mobile phone, filming.

The following 48 terms included in Appendix A regarding specific forms of school bullying, also mentioned in the text, are not clear and should be explained, such as: block, cyber, head, image, mark, targeted, video-clip.

Moreover, the information regarding the age of students in the South Korea, pupils who attend the schools and are exposed to the  risk of school violence would be needed.

Regarding the title and content of the manuscript, in the literature the wording of “school bullying forms”  is more frequent used than “school bullying methods”.

In Introduction part reference [2] (line: 34) seems to be not corrected, compering to the List of References https://doi.org/10.1016/j.futures.2007.08.021. Also not sure if ref. [45] is adequate  refereed to the content (line:350). In Discussion referring to female students is not appropriate since the study do not analyses genders (line:304)

In Results: Figure 3 and Figure 4 are illegible,  not possible to read.

Author Response

Response to Reviewer #2

1.     The description of the selected forms of school bullying forms is not sufficient. It would be helpful to explain more the forms (methods) not related by meaning to the school bullying. In Table 2 the forms (methods) of school bullying should be better described as related to school bullying, such as: video-clip, head, text message, cyber, image, mobile phone, filming. The following 48 terms included in Appendix A regarding specific forms of school bullying, also mentioned in the text, are not clear and should be explained, such as: block, cyber, head, image, mark, targeted, video-clip.

è The authors thank the Reviewer for making this important suggestion. We have provided more information about the words of school bullying forms mentioned in the manuscript in Table 1 and Appendix A.

(p.7, 11) Most words directly describe forms of school bullying, however, some neutral words indicate the tools used in bullying events (e.g., blanket, image, mobile phone, text message, and video-clip); some indicate exclusion behavior in SNS (e.g., block, block list); some indicate teasing behavior using digital devices (e.g., capture, filming); some indicate the result of bullying behavior (e.g., mark); others are detailed description of bullying behavior (e.g., cyber, targeted).

2.     Moreover, the information regarding the age of students in the South Korea, pupils who attend the schools and are exposed to the risk of school violence would be needed.

è Thank you for your suggestion. The authors have added more information about the age of students in South Korea in the Introduction section.

(p.1) As stated in a recent national report, the most common forms of bullying victimization in Korea among elementary school to high school students (primarily between the ages of 7 and 18) were verbal aggression (34.7%) and group ostracism (17.2%), followed by stalking (11.8%), cyber (10.8%) and physical (10.0%) bullying [1].

3.     Regarding the title and content of the manuscript, in the literature the wording of “school bullying forms” is more frequent used than “school bullying methods”.

è The authors appreciate the Reviewer’s suggestion. In the entire manuscript, we have changed the wording of “school bullying methods” to “school bullying forms”.

4.     In Introduction part reference [2] (line: 34) seems to be not corrected, compering to the List of References https://doi.org/10.1016/j.futures.2007.08.021. Also not sure if ref. [45] is adequate refereed to the content (line:350).

è The authors appreciate the detailed review of the manuscript. We have corrected reference [3] (line: 37, 400; which was ref. [2] in original manuscript) and removed reference [45] (line: 349, 517).

5.     In Discussion referring to female students is not appropriate since the study do not analyses genders (line:304)

è While the authors agree that analyzing gender differences is not a key to the current study goals, we would like to include comments concerning female students in the Discussion section as the recent surge of violent female bullying involvement is unique to the Korean context. Instead, we have modified the original sentence for clarification.

(p.9) The recent surge in group assault incidence by adolescents and physical violence cases particularly among female students in Korea has also been indicated using traditional social science research methods [16, 36].

6.     In Results: Figure 3 and Figure 4 are illegible, not possible to read.

è The authors thank the Reviewer for this suggestion. We have removed Figures 3 and 4.